# Organic Waste Gasification by Ultra-Superheated Steam

**Sergey M. Frolov**

Department of Combustion and Explosion, Semenov Federal Research Center for Chemical Physics of the Russian Academy of Sciences, Moscow 119991, Russia; smfrol@chph.ras.ru

**Abstract:** The perspective of the emerging environmentally friendly and economically efficient detonation gun technology for the high-temperature gasification of organic wastes with ultra-superheated mixture of steam and carbon dioxide is discussed. The technology is readily scalable and allows the establishment of a highly reactive atmospheric-pressure environment in a compact water-cooled gasifier due to very high local temperature (above 2000 °C), intense in situ shock-induced fragmentation of feedstock, and high-speed vortical convective flows enhancing interphase exchange processes. These unique and distinctive features of the technology can potentially provide the complete conversion of solid and liquid wastes into syngas, consisting exclusively of hydrogen and carbon monoxide; microparticles, consisting of environmentally safe simple oxides and salts of mineral residues, as well as aqueous solutions of oxygen-free acids such as HCl, HF, $H_2S$, etc., and ammonia $NH_3$. A small part of the syngas product (ideally approximately 10%) can be used for replacing a starting fuel (e.g., natural gas) for the production of a detonation-born gasifying agent, while the rest can be utilized for the production of electricity, heat, and/or chemicals.

**Keywords:** organic waste; gasification; detonation; ultra-superheated steam; carbon dioxide; syngas

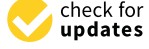



## 1. Introduction

The thermal processing of liquid/solid organic wastes by hot water/steam (hydrothermal gasification [1,2], supercritical water gasification [3,4], steam gasification [5–9]) and carbon dioxide [10–12] is considered a competitive and economically viable technology for waste-to-energy [13–15], waste-to-hydrogen [16–20], waste-to-fuel [21,22], and waste-to-plastic [23–25] production, especially when the heat required for processing is obtained by environmentally clean technologies (plasma [26–28], microwave [29,30], solar [31–33], etc.) different from feedstock combustion. The current status, challenges, and perspectives of the industrial thermal processing of organic wastes all over the world was recently reported in [34,35]. The use of steam and/or carbon dioxide as gasifying agents has several advantages [36,37]. Steam and $CO_2$ are composed only of H and O and C and O atoms; respectively, and therefore, the produced syngas is not diluted by other gases. Steam/$CO_2$ gasification of waste requires a lower amount of the gasifying agent due to their high enthalpies. The use of a blended $H_2O/CO_2$ gasifying agent allows for manipulation of the syngas composition. The use of steam as a gasifying agent increases cost efficiency [38,39]. Finally, in the absence of free oxygen, the syngas produced by steam/$CO_2$ gasification does not contain such toxic compounds as dioxins and furans [40], which facilitates gas cleaning operations. The amount of hydrogen produced through the steam gasification of a biomass is approximately a factor of three larger than that of the air gasification of a biomass. The $CO_2$ gas for $CO_2$-assisted gasification of organic wastes can be taken from the flue gas of power plants, thus, decreasing greenhouse gas emissions and reducing the carbon footprint.

Depending on the level of the gasification temperature and the heat source for gasification, all gasification technologies are categorized into autothermal/allothermal and low/high temperature technologies [41]. The autothermal technologies use the heat produced in a gasifier due to the addition of oxygen or air for partial combustion of the

feedstock, while the amount of the feedstock to be burned can be significant [42,43]. The allothermal technologies use the heat for gasification introduced from an external source such as electric heaters, heat exchangers, plasma arcs, heat carriers, etc. [44].

Low-temperature gasification is commonly related to temperatures below 1000 °C and results in the production of syngas, char, and slag. The syngas is usually contaminated with tar, $CO_2$, and particulate matter, as well as complex alkali metal, chlorine, and sulfur compounds. The removal of the various impurities from the syngas is, thus, a necessary step of the low-temperature gasification technologies. The char is the solid residue of organic wastes. It is composed mainly of carbon but can contain some hydrogen and oxygen, as well as inorganic ash. The slag is a glass-like byproduct of gasifiers that is considered nonhazardous and can be used as roofing material or in roadbed construction, etc. The main drawbacks of the existing low-temperature steam/$CO_2$ gasification technologies consist of low-quality syngas with high contents of tar and $CO_2$, low gasification efficiencies due to high char residues, complicated in situ control of syngas quality caused by long residence times of feedstock in gasifiers, and low yields of syngas caused mainly by its partial use for the production of heat for gasification. The current research and development efforts are mainly directed at feedstock preprocessing and improving its reactivity [45].

High-temperature gasification is related to temperatures above 1200 °C. Such temperatures are typical for combustion-assisted heating systems, as well as plasma and solar gasifiers. At such temperatures, the products of gasification are high-quality syngas and slag, as the syngas is composed mainly of $H_2$ and CO, whereas contents of hydrocarbons higher than $C_1$-$C_2$ are negligible, and alkali metal, chlorine, and sulfur compounds take on simple chemical structures. The main advantages of high-temperature steam/$CO_2$ gasification technologies consist of high-quality syngas due to negligible or low content of tar and $CO_2$, high gasification efficiencies due to small tar and char residues, easy in situ control of syngas quality due to short residence times of feedstock in a gasifier, and high yields of syngas due to the use of external energy sources for the production of heat for gasification. Despite these advantages, there are also certain drawbacks inherent in the high-temperature plasma and solar gasification technologies that limit their widespread application. Industrial scale arc/microwave plasma technologies require tremendous electric power due to the gas–plasma transition, which looks unnecessary as the typical operation temperature of plasma gasifiers is approximately 1300–1700 °C. The lifetime of arc electrodes is limited. Due to high operation temperatures, the gasifier walls must be water cooled and made of special construction materials with refractory liners. The gasification efficiency of microwave plasma depends on feedstock properties. The intermittent character of solar-assisted gasification is its main drawback.

In view of all this, the emerging high-temperature technology of organic waste gasification by the ultra-superheated steam (USS) is of particular interest. The USS is the steam heated to temperatures above 1500–2000 °C. The ability of USS to gasify liquid/solid organic wastes producing no negative effects to the environment is well known [41]. At temperatures above 1500 °C, tar and char formed at the initial stages of the gasification process are completely transformed to syngas, ideally composed only of hydrogen and carbon monoxide in a proportion depending on the feedstock, whereas condensed mineral residues consist of safe simple oxides and aqueous solutions of oxygen-free acids such as HCl, HF, $H_2S$, etc., and ammonia $NH_3$ [46–48]. The mineral residues can be used as additives in building materials, while acids can be separated, concentrated, and placed on the market. In other words, organic waste gasification by the USS potentially allows its complete conversion into useful products without emissions into the atmosphere and water bodies.

## 2. How to Produce the Ultra-Superheated Steam

Conventional steam generators (boilers) produce steam in heat-exchanger water tubes. The heat is usually supplied by burning gas, oil, or coal. The maximum steam temperature in such boilers does not exceed ~1000 °C because of limitations on the thermal resistance of

heat exchanger materials. This means that conventional steam generators cannot be used for producing the USS.

For producing the USS, a new method is proposed [41]. The essence of the method is illustrated in Figure 1. The main element of the USS generator is a pulsed detonation gun (PDG). The PDG is simply a tube with one closed and another open end. The closed end is equipped with ports for the supply of fuel, oxygen, and low-temperature steam (110–120 °C) from the corresponding manifolds with control valves. The operation cycle starts with partially filling the PDG with a plug of low-temperature steam with length $L$ ($L$ is determined by the steam flow rate and fill time), followed by supplying the components of a reactive mixture: fuel, oxygen, and in general, low-temperature steam as a diluent. After ignition of the reactive mixture with a spark igniter (see Figure 1a), flame acceleration, and deflagration-to-detonation transition (DDT), a developed detonation wave propagates at a very high velocity (~2000 m/s) through the reactive mixture, transforming it into the detonation products, composed primarily of USS and $CO_2$ at very high temperature and pressure (see Figure 1b). When transitioning into the plug of low-temperature steam, the detonation wave is converted into a strong but decaying shock wave that compresses and heats up the steam to a very high temperature and pressure (see Figure 1c). After the shock wave exits through the open end of the PDG, the high-temperature and high-pressure USS begins to expand into the ambience as a high-velocity (in average, above 1000 m/s) jet (see Figure 1d). When the pressure in the PDG drops down to the ambient pressure, a new plug of low-temperature steam is supplied to the PDG through the port at the closed end (see Figure 1e), followed by a feed of the reactive mixture components: fuel, oxygen, and low-temperature steam as a possible diluent (see Figure 1f). Upon completely filling the PDG, a spark igniter is triggered to begin the next operation cycle (see Figure 1a). Note that the detonation products of the mixtures of fuel with oxygen can contain enough USS and $CO_2$ that the dilution of the reactive mixture with the low-temperature steam can be optional.

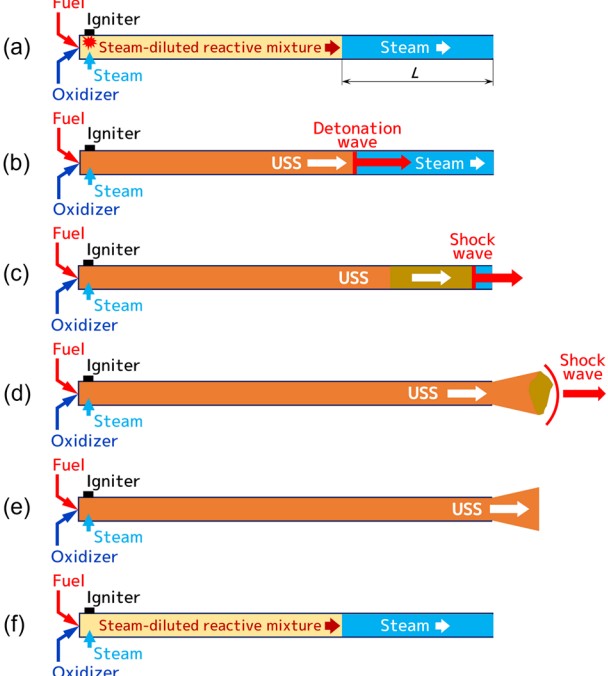

**Figure 1.** Phases of the operation cycle of the USS generator: (**a**) ignition of the reactive mixture; (**b**) flame acceleration and DDT followed by detonation propagation; (**c**) detonation-to-shock transition followed by shock compression and heating of steam; (**d**) USS expansion to ambience; (**e**) filling with steam; (**f**) filling with the reactive mixture. The operation cycle continues by repeating phase (**a**).

The USS generator operates in a pulsed mode with the pulse frequency determined by the PDG fill time, as the fill time is an order of magnitude longer than the total time taken for flame ignition and acceleration, DDT, and detonation propagation along the PDG. During the fill process, the reactive mixture should not come into contact with the USS under any circumstances, which is guaranteed by the use of the plug of low-temperature steam. Otherwise, the reactive mixture can be ignited by the USS, leading to the violation of the operation cycle. It is also implied that only USS can be issued through the open end of the PDG, i.e., neither the low-temperature steam nor the reactive mixture can issue through it without being affected by the detonation or shock wave. The average parameters of the issuing USS jet (temperature, composition, velocity, etc.) are determined by the reactive mixture composition, the degree of its dilution with the low-temperature steam, and the length of the low-temperature steam plug (*L*).

The USS generator under discussion exhibits several important advantages. First, it can be made of conventional construction materials because the PDG is effectively cooled down from the interior during the fill process and can be water cooled from the exterior with further use of the heated water for producing the low-temperature steam required for the operation process. Second, it is able to operate on any available gaseous (natural gas, propane–butane, syngas, biogas, etc.) or liquid (gasoline; diesel oil; pyrolysis oil; methanol; oil-in-water emulsions; hydrous compositions, such as binary coal–water suspensions, triple organic–coal–water suspensions; and anhydrous compositions such as mixtures of motor fuels or biofuels with solid particles, etc.) hydrocarbon fuel, as such fuel–oxygen mixtures usually exhibit high detonability in the wide range of fuel content and steam/water dilution. For example, stoichiometric natural gas–oxygen and propane–oxygen mixtures support detonation propagation even when diluted by steam up to 40 and 60%vol, respectively [49]. Furthermore, it can effectively operate on the syngas produced by gasification of the various organic wastes. The latter is possible if the syngas product is not too diluted with inert gases. Third, due to its simple design and operation principle, it can be easily scaled up to industrial-scale USS flow rates using the known scaling criteria of gaseous or heterogeneous detonations. Interestingly, when scaling the USS generator, one can replace the straight PDG with a bent or coiled PDG without significantly affecting the operation process, which is caused by a highly localized reaction zone in a propagating detonation wave. The energy consumption for the cyclic detonation ignition is negligible.

In addition to the mentioned advantages, the USS generator has several limitations. The first one relates to the use of oxygen or air enriched with oxygen as an oxidizer for the PDG operation. If air rather than oxygen is fed to the PDG, then the syngas product will be highly diluted with nitrogen. In this case, the DDT in the syngas–air mixture will be hardly possible without adding extra oxygen to the air. The second one relates to the necessity of using fast-acting check valves on the fuel, oxygen, and steam supply lines to prevent the flame from flashing up the lines.

Another option for the USS generator based on the rotating detonation gun (RDG) is also worth noting [41,50]. The RDG operates on the detonation wave continuously rotating in an annular gap and exhibits a USS production rate 2–3 orders of magnitude higher than that typical of the PDG.

## 3. How to Gasify Organic Waste with the Ultra-Superheated Steam

The USS generator of Section 2 can be attached via the open end to the flow-through gasifier of compact geometrical configuration, which avoids the formation of long-lived flow stagnation zones promoting waste accumulation and slagging inside the gasifier. The flow-through gasifier can be water cooled with the heated water further used for producing the low-temperature steam required for the operation process. USS generators must be preferably attached to the gasifier in pairs and must be located coaxially opposite each other to generate strong colliding shock waves and powerful vortical structures aimed at increasing the residence time of waste particles and their carbonized residues inside the

gasifier (Figure 2a). Pulsed shock waves emanating from the USS generators, having a huge destructive force, grind waste into small particles, as well as prevent particle agglomeration and particle adhesion to the walls of the gasifier. Importantly, waste particles can undergo multiple acts of fragmentation by the successive incident and reflected shock waves, as well as undergo repeated involvement in the high-temperature vortical structures of the USS away from the relatively cold walls during their stay in the gasifier. The gasifier must also include an inlet for continuously feeding the organic waste, an outlet for continuous venting of the syngas product, and an outlet for continuous or batch removal of mineral residues. To increase the mean residence time of waste particles, one can use the cascade of gasifiers communicating with each other via the tubes, allowing gases and particles to flow from one gasifier to another (Figure 2b). In all cases, the average operation pressure in the gasifier (s) must be slightly above the atmospheric pressure to avoid the suction of atmospheric air. At present, such a USS gasification plant meeting specific requirements in terms of the operation temperature, feedstock residence time, syngas product composition, etc., can be designed using various available computational fluid dynamics (CFD) approaches [51–53] and equilibrium gasification models [54,55].

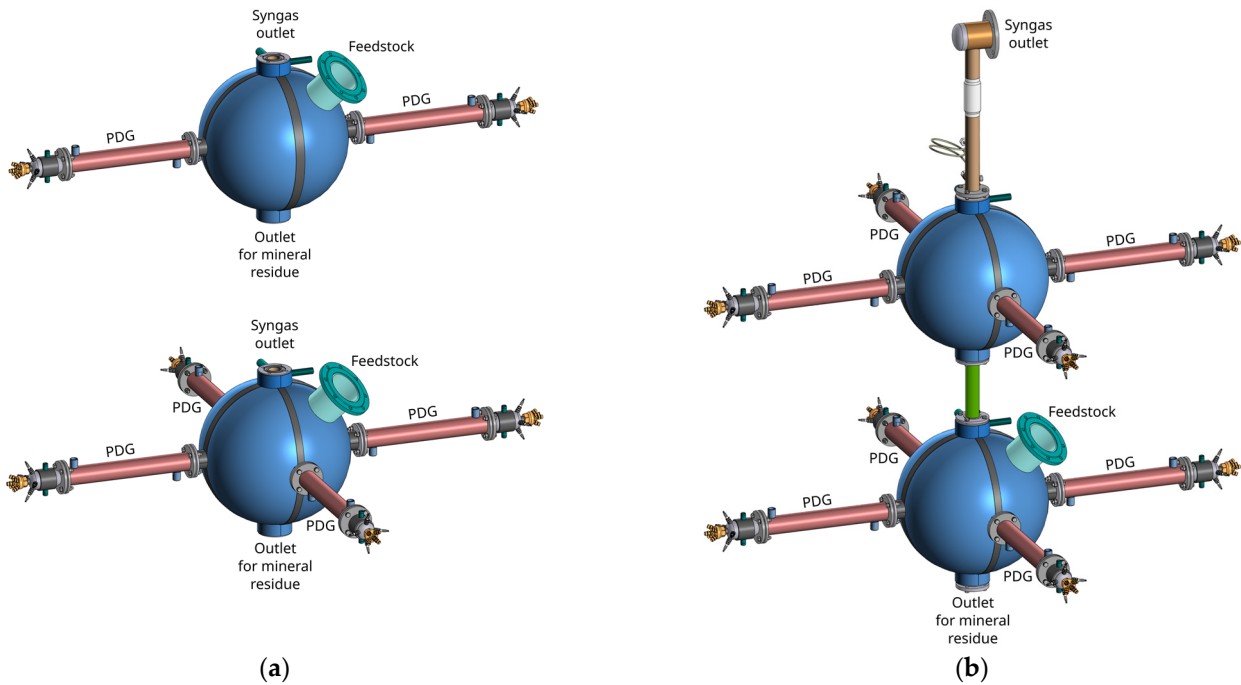

(**a**)  (**b**)

**Figure 2.** Schematics of the units for organic waste gasification by the USS: (**a**) a single flow-through gasifier with one and two pairs of PDGs; (**b**) a cascade of two flow-through gasifiers with two pairs of PDSs.

The operation of the gasification plant goes through three transient stages until the nominal operation mode is reached. The goal of the first transient stage is to bring the plant to the stationary mode of operation in terms of the steady thermal state of all its elements and cooling water. At this stage, a feedstock is not supplied to the gasifier. After the steady thermal state is reached, the second transient stage comes into play, which is when feedstock begins to be continuously fed into the gasifier. In some time, another stationary mode of operation is established in terms of the new steady thermal state of all its elements and cooling water, as well as the steady-state composition of the syngas product. During the first and second transient stages, the plant operates on an available starting fuel, e.g., natural gas. The goal of the third transient stage is to gradually replace the starting fuel with the product syngas and bring the plant to the steady-state nominal mode of operation. In this mode, the composition of the product syngas is optimal in terms of the volume fractions of $H_2$ and CO, the major components of the syngas, and the

volume fraction of the residual steam required for the downstream condensation of acids and ammonia in the form of mixed concentrated aqueous solutions. Due to the extremely high temperatures of the USS entering the gasifier from the PDGs and the sufficiently long mean residence time of feedstock particles in the gasifier (s), the volume fractions of $CO_2$, $CH_4$, and other hydrocarbons must tend to zero [41]. Secondary pollutants such as microparticles of metal oxides and salts can be separated from the syngas product by extraction with a proper acid solution, followed by the transformation of metal salts to safe carbonates. As a result, the cooled syngas product must be of very high quality, containing only $H_2$ and CO and requiring no further cleaning. At the nominal operation mode, only a part of the cooled product syngas is used as a fuel. For this purpose, this part of the syngas must be compressed to ensure the required mass flow rate through the PDGs.

One of the most important limitations in the described operation of the gasification plant relates to the continuous feed of the organic waste in the form of the loose solid material. The conventional screw feeders are not directly applicable because the shock waves emanating from the PDGs can compress the loose material, whereas the USS can penetrate into the material and coke it. As a result, the screw feeder can experience difficulties in operation. The good news is that there are no problems with the feed of liquid and pasty wastes.

The most important systems in the gasifier are safety and control systems [48]. During normal operation of the gasifier, flame arresters with fast-acting check valves on the fuel, oxygen, and steam supply lines must prevent the flame from flashing up the lines. A misfire or a failure of DDT in the PDG can result in a release of explosive mixture to the flow reactor with the danger of subsequent internal explosion. The danger of internal explosion is increased by several successive misfires or failures of DDT. Therefore, the gasifier must withstand an accidental internal explosion with the maximum overpressure inherent in the constant-volume explosion of the most energetic fuel–oxygen mixture used at any stage of the operation process. To improve ignition reliability, several spark plugs must be installed in the PDGs. In addition, a pressure sensor installed at the reactor inlet must be used to monitor a misfire: if the overpressure is below a certain threshold value, the PDG must be purged with low-temperature steam. To ensure the inherent safety of the operation process in case of DDT failure, the PDGs must always be filled only partly with the explosive mixture. When DDT fails, the flame front "covers" the entire charge of the explosive mixture before it enters the gasifier. In addition, to monitor the failure of DDT, the ion sensors must be installed in the PDGs. Based on the readings of the sensors, the instantaneous propagation velocities of detonation waves must be monitored. When DDT fails, the PDG must be purged with low-temperature steam. There is also a need to monitor the pressure and temperature at the gasifier outlet using a pressure sensor and thermocouple. Thermocouples are also used to monitor the coolant temperature in the water-cooling jackets of the PDGs and gasifier.

As an example, Figure 3 shows the flow chart of the gasification plant operation. It is assumed that the plant operates on the oxygen mixture of the syngas produced by the gasification of a specific organic feedstock, which provides syngas with a molar $H_2$/CO ratio of 1.5. The concentrations of the major species in the boxes are calculated using the thermodynamic code (see [41,47,48]). Box #1 corresponds to the feed of syngas and oxygen in the stoichiometric amounts, as well as low-temperature steam as a diluent to the PDGs. Box #2 shows the approximate composition, temperature, and pressure of the detonation products in the PDGs. Box #3 shows the approximate composition and temperature of the detonation products expanded from the PDGs to the gasifier operating at a mean pressure slightly above 1 bar. Box #4 shows the gasifier with the supply of a feedstock and gasifying agent, as well as removal of the hot product syngas. Box #5 shows the expected composition of the cooled dry syngas. Approximately 10%wt. (see the next section) of the cooled dry syngas is compressed to approximately 5 bar and directed back to the USS generator through the compressor (box #6). Finally, approximately 90%wt. of the cooled dry syngas is directed to the user.

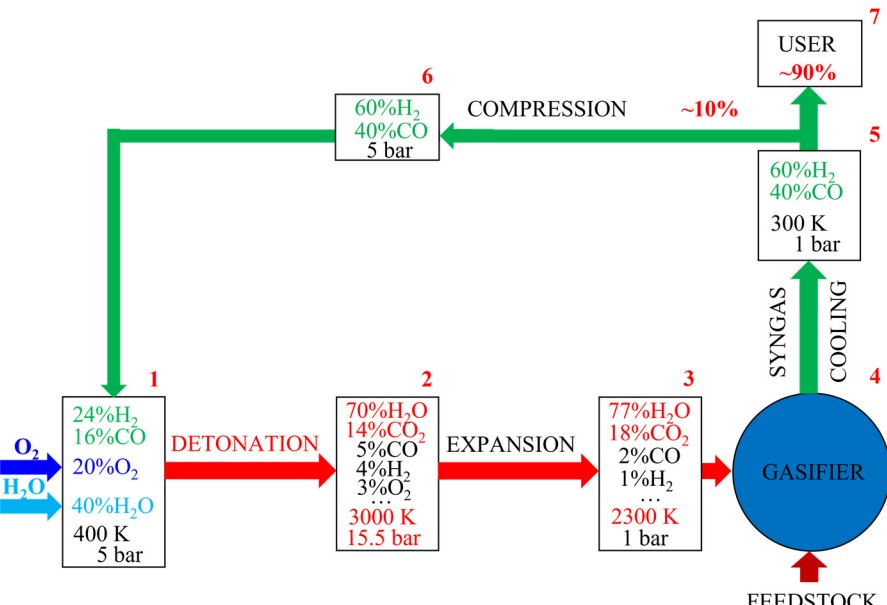

**Figure 3.** Flow chart of waste gasification by ultra-superheated steam.

Recent experimental studies [47,48] confirm the high quality of the syngas product obtained from natural gas, liquid (waste machine oil, heating value ~42 MJ/kg), and solid (birch wood sawdust, heating value ~18 MJ/kg) wastes by the detonation-born USS conversion/gasification in the gasifiers 40 and 100 L in volume. Figure 4 shows the averaged measured compositions of the cooled dry syngas in terms of the dependences of the steady-state values of $H_2$, CO, and $C_xH_y$ volume fractions on the $CO_2$ volume fraction. The scatter of the experimental data relevant to Figure 4 is approximately 5%vol. for $H_2$, CO, and $CO_2$ and 2%vol. for $C_xH_y$. The $C_xH_y$ is mainly represented by $CH_4$ (hydrocarbons larger than $C_2$ were not found in the gas probes [47,48]). With the increase in the process temperature, the volume fraction of $CO_2$ decreases, leading to a gradual increase in the volume fractions of $H_2$ and CO to approximately 60 and 40%vol dry basis (d.b.), whereas the volume fraction of $C_xH_y$ reaches a maximum at a $CO_2$ volume fraction at approximately 10–20% d.b. and tending to zero at $CO_2 \rightarrow 0$.

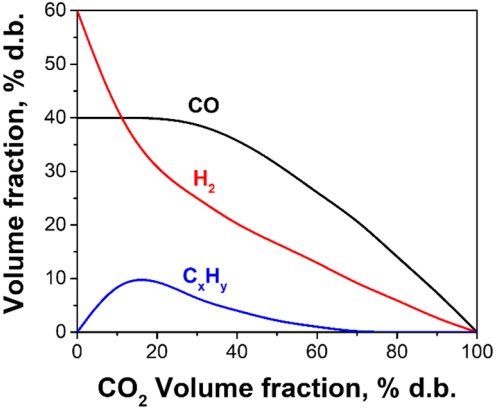

**Figure 4.** Measured syngas compositions in experiments with natural gas conversion and liquid/solid waste gasification by the detonation-born USS.

## 4. Economic Considerations

The ideal mass and energy balances for the technology under discussion were calculated in [47] for the dry wood sawdust as a feedstock. In the calculations, the PDG was assumed to operate on the steam diluted (40%vol. $H_2O$) stoichiometric mixture of syngas

($H_2$/CO = 1) with $O_2$. The syngas of such a composition was assumed to be produced as a result of the full gasification of the biomass in the gasifier. It was also assumed that the thermal energy of the detonation products was completely recuperated by the cooling system to the energy of the low-temperature steam entering the PDG as a diluent, i.e., irreversible losses were absent. The calculations included the energy for $O_2$ production (~4 MJ/kg), which was ~12 MJ/kg with regard for production efficiency (33%); the energy required for the compression of cooled dry syngas from 1 to 5 bar before supply to the PDG (~0.3 MJ/kg), which was ~0.9 MJ/kg with regard for compression efficiency (33%); and the heat of the formation of dry wood sawdust (in the range between –5.7 and –4.9 MJ/kg [56]. It was shown that for gasifying 1 ton of dry sawdust, ideally, 145 kg syngas, 155 kg $O_2$, and 175 kg water are needed. As a result of full gasification, 1000 + 145 + 155 + 175 = 1475 kg syngas was obtained, a part of which (145 kg, i.e., ~10%) was used for the gasification of the next one ton of sawdust. The energy gain defined as the ratio of the total energy of the syngas leaving the gasifier to the energy spent in its production was ~4.6. When biomass moisture was included (10%wt), a part of the energy had to be spent on sawdust water vaporization. In this case, the mass of the processed wet sawdust was ~13% less, while the total mass of the product syngas decreased by 9%. Extra steam (10%wt) was condensed and directed back to the USS generator.

When applied to the syngas with $H_2$/CO = 1.5 and 2, the calculations of the mass and energy balances led to results that were quite similar to those reported in [47]: in the nominal operation mode, the gasification of dry biomass required the removal of approximately 10%wt of the product syngas for its own needs, which resulted in an energy gain on the level of 5–6. Such an energy gain was recently confirmed experimentally [57], which means that the technology under discussion is potentially very energy efficient and can be considered a feasible way of organic waste utilization [58]. Notably, electricity is needed only for the operation of the low-flow water pump and the low-pressure syngas compressor, as well as the steam generator for obtaining slightly superheated steam (110–120 °C). The energy consumption for detonation ignition (on the order of a fraction of a Joule per cycle) and control system operation is very small.

## 5. Conclusions

The perspective of the emerging environmentally friendly and economically efficient technology for the high-temperature oxygen-free allothermal gasification of organic wastes with detonation-born ultra-superheated mixture of steam and carbon dioxide was discussed. The technology is implemented in a very simple plant containing a compact water-cooled flow-through gasifier and several pulsed detonation guns, preferably attached to the gasifier coaxially opposite each other. The detonation guns periodically generate strong colliding shock waves and jets of an ultra-superheated mixture of steam and carbon dioxide and create powerful vortical structures in the gasifier, ensuring a long residence time of waste particles and their carbonized residues inside the gasifier.

The technology has several attractive features. It allows operation at atmospheric pressure and extremely high gasification temperatures obtained by detonating a part of the produced syngas (approximately 10%) as a process fuel, while the energy consumption for detonation ignition is negligible. It allows the complete conversion of gas, liquid, and solid wastes into useful products, namely, high-quality syngas, fine particles of mineral residues, and aqueous solutions of simple acids. It can be implemented using conventional structural materials and is readily scaled-up. Finally, it is economically beneficial, as the ideal energy gain for syngas production is approximately a factor of 5 to 6.

**Funding:** This research received no external funding.

**Data Availability Statement:** The data is available by request.

**Conflicts of Interest:** The author declares no conflict of interest.

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
