# Peer review of "Organic Waste Gasification by Ultra-Superheated Steam"

_energies, doi:10.3390/en16010219_

Round 1

Reviewer 1 Report

The author provided a technical perspective on the gasification of organic waste by ultra superheated steam (USS). It is very topical in the era of plastic and other organic wastes blighting the planet and causing climate change and having other harmful effects on the environment. The method identified by the author for USS generation is by controlled explosions using pulsed detonation gun and steam of up to 1500 oC can be achieved. It is noted that this method has syngas as byproduct, which is itself a fuel. 

the paper is well-written, and can be published. The reviewer only has minor suggestions for improvement after which the paper may be accepted for publication. 

1. since the work is composed of review of the literature, references are needed for the figures in the manuscript for figs. 1 - 4. 

2. How is the length L in fig. 1 determined? What are the factors affecting what this length will be?

3. No mention was made of the heating values of the possible wastes (for fuels) that can be used for the PDG process

4. i think it will be good to tabulate or plot a bar chart of the economics of using this method compared to other methods. Definitely the quantitative data in section 4 should be presented in infographics. 

5. Last sentence in the conclusion: "Finally, it is economically 297 beneficial, as the ideal energy gain for syngas production is around 5–6." - what is the unit of 5-6 or do you meant 5-6 times?

6. The conclusion needs to be made stronger to summarise the entire work, right now it is too short and rather light in detail. 

Author Response

I am grateful to the reviewer for valuable comments. I have made my best to follow all the comments. All changes in the revised manuscript are marked in yellow.

The author provided a technical perspective on the gasification of organic waste by ultra superheated steam (USS). It is very topical in the era of plastic and other organic wastes blighting the planet and causing climate change and having other harmful effects on the environment. The method identified by the author for USS generation is by controlled explosions using pulsed detonation gun and steam of up to 1500 oC can be achieved. It is noted that this method has syngas as byproduct, which is itself a fuel.

the paper is well-written, and can be published. The reviewer only has minor suggestions for improvement after which the paper may be accepted for publication.

  1. since the work is composed of review of the literature, references are needed for the figures in the manuscript for figs. 1 - 4.

Actually, these figures I have plotted exclusively for this Perspective article to explain the issues addressed. They were not published anywhere so far.

  1. How is the length L in fig. 1 determined? What are the factors affecting what this length will be?

The length L is determined by the steam flow rate and fill time. I have added this explanation to the text.

  1. No mention was made of the heating values of the possible wastes (for fuels) that can be used for the PDG process

I have added the heating values of waste machine oil and birch wood sawdust, when discussing the results of experiments in Section 3:

“(waste machine oil, heating value ~42 MJ/kg), and solid (birch wood sawdust, heating value ~18 MJ/kg)”

  1. i think it will be good to tabulate or plot a bar chart of the economics of using this method compared to other methods. Definitely the quantitative data in section 4 should be presented in infographics.

The elementary economic calculations were reported in detail in Ref. [49], where a big table is presented with all numbers. This Reference is cited just at the beginning of Section 4. The interested reader can readily repeat those elementary calculations for the feedstock of any origin. For this purpose, I give another reference (Ref. [57]), which helps one to estimate the heat of formation of any feedstock with the known atomic composition.

  1. Last sentence in the conclusion: "Finally, it is economically beneficial, as the ideal energy gain for syngas production is around 5–6." - what is the unit of 5-6 or do you meant 5-6 times?

Yes, I mean 5-6 times. I have replaced it by “ a factor of 5 to 6.”

  1. The conclusion needs to be made stronger to summarise the entire work, right now it is too short and rather light in detail.

I have added a short description of the gasification plant to the Conclusion section:

“The technology is implemented in a very simple plant containing a compact water-cooled flow-through gasifier and several pulsed detonation tubes preferably attached to the gasifier coaxially opposite each other. The detonation tubes periodically generate strong colliding shock waves and jets of ultra-superheated mixture of steam and carbon dioxide and create powerful vortical structures in the gasifier ensuring long residence time of waste particles and their carbonized residues inside the gasifier.”

Reviewer 2 Report

The present perspective article is aimed to summarize the “organic waste gasification by ultra-superheated steam (USS)”. The contents like, how to produce the ultra-super-heated system and how to gasify organic waste by the ultra-superheated steam along with the economic considerations were discussed. In the abstract only it is mentioned that the USS needed extremely high temperatures (above 2000 °C) to provide the intense in situ shock-induced fragmentation of feedstock. Also, the mentioned that it is consisting of environmentally safe simple oxides and salts, along with the acids such as HCl, HF, and H2S. There are safety and environmental complications associated with this process, authors highlighted the process by ignoring its limitations and secondary pollutants information. Detailed information including practical applicability along with the energy requirements, secondary pollutants, temperature maintenance, etc. needs to be discussed elaboratively.

Author Response

I am grateful to the reviewer for valuable comments. I have made my best to follow all the comments. All changes in the revised manuscript are marked in green.

The present perspective article is aimed to summarize the “organic waste gasification by ultra-superheated steam (USS)”. The contents like, how to produce the ultra-super-heated system and how to gasify organic waste by the ultra-superheated steam along with the economic considerations were discussed. In the abstract only it is mentioned that the USS needed extremely high temperatures (above 2000 °C) to provide the intense in situ shock-induced fragmentation of feedstock. Also, the mentioned that it is consisting of environmentally safe simple oxides and salts, along with the acids such as HCl, HF, and H2S.

There are safety and environmental complications associated with this process, authors highlighted the process by ignoring its limitations and secondary pollutants information. Detailed information including practical applicability along with the energy requirements, secondary pollutants, temperature maintenance, etc. needs to be discussed elaboratively.

Regarding technology limitations, one of them was already mentioned in the original text in Section 2:

“Thus, if air rather than oxygen is fed to the PDG as an oxidizer, then the product syngas will be highly diluted with nitrogen. In this case, the DDT in the syngas–air mixture will be hardly possible without adding extra oxygen to the air.”

In view of this comment, I have moved the above statement to a separate paragraph in Section 2 and added another important limitation:

“In addition to the mentioned advantages, the USS generator has several limitations. The first one relates to the use of oxygen or air enriched with oxygen as an oxidizer for PDG operation. If air rather than oxygen is fed to the PDG then the product syngas will be highly diluted with nitrogen. In this case, the DDT in the syngas–air mixture will be hardly possible without adding extra oxygen to the air. The second one relates to the necessity of using fast-acting check valves on the fuel, oxygen, and steam supply lines to prevent the flame from flashing up the lines.”

One more limitation is mentioned further in the text in Section 3:

“One of the most important limitations in the described operation of the gasification plant relates to the continuous feed of the organic waste in the form of the loose solid material. The conventional screw feeders are not directly applicable because the shock waves emanating from the PDGs can compress the loose material, whereas the USS can penetrate into the material and coke it. As a result, the screw feeder can experience difficulties in operation. The good news is that there are no problems with the feed of liquid and pasty wastes.”

Regarding the secondary pollutions, I have added a sentence:

“Secondary pollutants like microparticles of metal oxides and salts can be separated from the product syngas by extraction with a proper acid solution followed by the transformation of metal salts to safe carbonates.”

To address the safety issues, I have added a new paragraph to the text:

“The most important systems in the gasifier are safety and control systems [49]. During normal operation of the gasifier, flame arresters with fast-acting check valves on the fuel, oxygen, and steam supply lines must prevent the flame from flashing up the lines. A misfire or a failure of DDT in the pulsed detonation gun can result in a release of explosive mixture to the flow reactor with a danger of subsequent internal explosion. The danger of internal explosion is increased by several successive misfires or failures of DDT. Therefore, the gasifier must withstand an accidental internal explosion with the maximum overpressure inherent in the constant-volume explosion of the most energetic fuel–oxygen mixture used at any stage of the operation process. To improve ignition reliability, several spark plugs must be installed in the PDGs. In addition, a pressure sensor installed at the reactor inlet must be used to monitor a misfire: if the overpressure is below a certain threshold value, the PDG must be purged with low-temperature steam. To ensure the inherent safety of the operation process in case of DDT failure, the PDGs must always be filled only partly with the explosive mixture. When DDT fails, the flame front “covers” the entire charge of the explosive mixture before it enters the gasifier. In addition, to monitor the failure of DDT, the ion sensors must be installed in the PDGs. Based on the readings of the sensors, the instantaneous propagation velocities of detonation waves must be monitored. When DDT fails, the PDG must be purged with low-temperature steam. There is also a need to monitor the pressure and temperature at the gasifier outlet using a pressure sensor and thermocouple. Thermocouples are also used to monitor the coolant temperature in the water-cooling jackets of the PDGs and gasifier.”

To follow the comment on energy requirements, I have added a sentence in Section 4:

“Note that electricity is needed only for the operation of low-flow water pump and low-pressure syngas compressor, as well as steam generator for obtaining slightly superheated steam (110–120 °C). The energy consumption for detonation ignition (on the order of a fraction of Joule per cycle) and control system operation is very small.”

As for the comment on the temperature maintenance, the original manuscript contains this information in Section 3:

“The operation of the gasification plant goes through three transient stages until the nominal operation mode is reached. The goal of the first transient stage is to bring the plant to the stationary mode of operation in terms of the steady thermal state of all its elements and cooling water. At this stage, a feedstock is not supplied to the gasifier. After the steady thermal state is reached, the second transient stage comes into play, when a feedstock starts to be continuously fed into the gasifier. In some time, another stationary mode of operation is established in terms of the new steady thermal state of all its elements and cooling water, as well as the steady-state composition of the product syngas. During the first and second transient stages, the plant operates on an available starting fuel, e.g., natural gas. The goal of the third transient stage is to gradually replace the starting fuel by the product syngas and bring the plant to the steady-state nominal mode of operation.”

Reviewer 3 Report

This paper discussed organic waste gasification by ultra-superheated steam. The method to produce the ultra-superheated steam, the organic waste gasification approach, and economic considerations are presented. The authors should consider the following comments to improve the paper. 

(1) Too many works of literature are cited without comments in the first sentence. Relevant studies should be reviewed in detail. The introduction should be revised. 

(2) The typical experiments of organic waste gasification by ultra-superheated steam should be introduced in detail. For present paper presents too much qualitative information. 

(3) The uncertainty should be discussed.

(4) The safety issue of high temperature and high pressure should also be discussed. 

Author Response

I am grateful to the reviewer for valuable comments. I have made my best to follow all the comments. All changes in the revised manuscript are marked in blue.

This paper discussed organic waste gasification by ultra-superheated steam. The method to produce the ultra-superheated steam, the organic waste gasification approach, and economic considerations are presented. The authors should consider the following comments to improve the paper. 

1. Too many works of literature are cited without comments in the first sentence. Relevant studies should be reviewed in detail. The introduction should be revised.

This manuscript is not intended for the review of all existing approaches. Such a comprehensive review, which is nearly 100 pages long, I have recently published in Fuels (see my Ref. 42). The idea was to discuss the new emerging technology of organic waste gasification by the ultra-superheated steam. The first sentence in the Introduction is just a list of the various technologies available to help the reader navigate.

2. The typical experiments of organic waste gasification by ultra-superheated steam should be introduced in detail. For present paper presents too much qualitative information.

In Sections 3 and 4 I provided such an example illustrated by Figures 3 and 4 with quantitative information, which generalizes the detailed experimental studies published in Refs. 48 and 49.

3. The uncertainty should be discussed.

To address this comment, I have added the following sentence to Section 3:

“The scatter of the experimental data relevant to Figure 4 is about 5%vol. for H2, CO, and CO2, and 2%vol. for CxHy.”

4. The safety issue of high temperature and high pressure should also be discussed.

I have added the paragraph in Section 3, which discusses the safety issues:

“The most important systems in the gasifier are safety and control systems [49]. During normal operation of the gasifier, flame arresters with fast-acting check valves on the fuel, oxygen, and steam supply lines must prevent the flame from flashing up the lines. A misfire or a failure of DDT in the pulsed detonation gun can result in a release of explosive mixture to the flow reactor with a danger of subsequent internal explosion. The danger of internal explosion is increased by several successive misfires or failures of DDT. Therefore, the gasifier must withstand an accidental internal explosion with the maximum overpressure inherent in the constant-volume explosion of the most energetic fuel–oxygen mixture used at any stage of the operation process. To improve ignition reliability, several spark plugs must be installed in the PDGs. In addition, a pressure sensor installed at the reactor inlet must be used to monitor a misfire: if the overpressure is below a certain threshold value, the PDG must be purged with low-temperature steam. To ensure the inherent safety of the operation process in case of DDT failure, the PDGs must always be filled only partly with the explosive mixture. When DDT fails, the flame front “covers” the entire charge of the explosive mixture before it enters the gasifier. In addition, to monitor the failure of DDT, the ion sensors must be installed in the PDGs. Based on the readings of the sensors, the instantaneous propagation velocities of detonation waves must be monitored. When DDT fails, the PDG must be purged with low-temperature steam. There is also a need to monitor the pressure and temperature at the gasifier outlet using a pressure sensor and thermocouple. Thermocouples are also used to monitor the coolant temperature in the water-cooling jackets of the PDGs and gasifier.”

Round 2

Reviewer 2 Report

The author considered the reviewer's comments and revised the manuscript accordingly, the present manuscript format is looking good.

Reviewer 3 Report

can be accepted